

# Can sea urchins beat the heat? Sea urchins, thermal tolerance and climate change

Elizabeth Sherman

Department of Natural Sciences, Bennington College, Bennington, VT, USA

## ABSTRACT

The massive die-off of the long-spined sea urchin, *Diadema antillarum*, a significant reef grazer, in the mid 1980s was followed by phase shifts from coral dominated to macroalgae dominated reefs in the Caribbean. While *Diadema* populations have recovered in some reefs with concomitant increases in coral cover, the additional threat of increasing temperatures due to global climate change has not been investigated in adult sea urchins. In this study, I measured acute thermal tolerance of *D. antillarum* and that of a sympatric sea urchin not associated with coral cover, *Echinometra lucunter*, over winter, spring, and summer, thus exposing them to substantial natural thermal variation. Animals were taken from the wild and placed in laboratory tanks in room temperature water ($\sim$22 °C) that was then heated at 0.16–0.3 °C min$^{-1}$ and the righting behavior of individual sea urchins was recorded. I measured both the temperature at which the animal could no longer right itself ($T_{\mathrm{LoR}}$) and the righting time at temperatures below the $T_{\mathrm{LoR}}$. In all seasons, *D. antillarum* exhibited a higher mean $T_{\mathrm{LoR}}$ than *E. lucunter*. The mean $T_{\mathrm{LoR}}$ of each species increased with increasing environmental temperature revealing that both species acclimatize to seasonal changes in temperatures. The righting times of *D. antillarum* were much shorter than those of *E. lucunter*. The longer relative spine length of *Diadema* compared to that of *Echinometra* may contribute to their shorter righting times, but does not explain their higher $T_{\mathrm{LoR}}$. The thermal safety margin (the difference between the mean collection temperature and the mean $T_{\mathrm{LoR}}$) was between 3.07–3.66 °C for *Echinometra* and 3.79–5.67 °C for *Diadema*. While these thermal safety margins exceed present day temperatures, they are modest compared to those of temperate marine invertebrates. If sea temperatures increase more rapidly than can be accommodated by the sea urchins (either by genetic adaptation, phenotypic plasticity, or both), this will have important consequences for the structure of coral reefs.

## INTRODUCTION

Adult sea urchins are grazers and have been dominant herbivores in many Caribbean reefs for decades and perhaps millennia (*Lessios, Garrido & Kessing, 2001*; *Levitan, Edmunds & Levitan, 2014*). A convincing and well-documented example of their influence on coral reef community structure occurred in the mid 1980's when Caribbean sea urchin

Corresponding author
Elizabeth Sherman,
sherman@bennington.edu

populations of *Diadema antillarum* experienced a massive die-off. The subsequent overgrowth of macroalgae resulted in decreased coral cover and recruitment and concomitant biodiversity losses from which reefs have not fully recovered (*Jackson, 2001*; *Levitan, Edmunds & Levitan, 2014*; *Miller et al., 2009*). Nevertheless, rebounds in *Diadema* populations in certain Caribbean locations are correlated with greater coral recruitment and coral growth (*Edmunds & Carpenter, 2001*; *Knowlton, 2001*; *Carpenter & Edmunds, 2006*; *Idjadi, Haring & Precht, 2010*), which might presage an increase in fish diversity (*Rogers, Blanchard & Mumby, 2014*). The proximate cause of the *Diadema* die-off was likely a disease agent that has never been identified (*Beck, Miller & Ebersole, 2014*), but additional environmental stressors such as increasing ocean temperature due to global climate change may also be deleterious for marine ectotherms such as echinoids particularly in intertidal and subtidal tropical habitats (*Nguyen et al., 2011*).

Tropical sea temperatures are predicted to increase by as much as 4.8 °C by the end of this century (*IPCC, 2014*). Threshold temperatures for coral bleaching have been evaluated in light of those predictions (e.g. *Tolleter et al., 2013*; *Hoegh-Guldberg, 1999*; *Donner, 2011*). There is also a growing literature on the effects of anticipated future temperature increases on fertilization and early development of echinoids (*Brennand et al., 2010*; *Byrne, 2011*; *Byrne et al., 2009*; *Sewell & Young, 1999*). The few previous studies of thermal tolerance in adult sea urchins, however, did not evaluate results in the context of global climate change (*Lawrence, 1973*; *Lawrence & Cowell, 1996*; *Ubaldo, Uy & Dy, 2007*). Moreover, no thermal tolerance data exist for the Caribbean keystone species, *Diadema antillarum.* Additionally, characterizing the thermal tolerance of sea urchins might permit predictions about changes in their geographic ranges (*Ling et al., 2009*).

I studied the thermal tolerance of the long-spined sea urchin, *Diadema antillarum*, and compared it to the sympatric rock-boring sea urchin, *Echinometra lucunter*, which, unlike *Diadema*, is not associated with low macroalgal cover (*Furman & Heck Jr, 2009*). My study site was in Grand Cayman, British West Indies, on the western side of the island within the Marine Protected Area (*McCoy, Dromard & Turner, 2009*). I collected urchins from depths ranging from just below the surface to 3 m. The animals were collected opportunistically based on their accessibility and whether they would be too large to accommodate in the lab or too small to be adults. Both species occurred at depths of 3 m, but only *Echinometra* occurred at the shallowest depths; I never found *Diadema* less than about 30 cm below the water surface. I would expect therefore that *Echinometra* might be exposed to higher daily temperatures than *Diadema* and thus might have a higher thermal tolerance (*Nguyen et al., 2011*). On the other hand, the rock boring behavior of *Echinometra* might mitigate temperature fluctuations given that they were always wedged into crevices during the day, while I found *Diadema* in exposed areas of the reef during the day.

I measured the thermal tolerance of individuals of these species over three seasons, and thus three different temperature regimes to determine if they acclimatized to changes in environmental water temperature. Finally, I measured variation in thermal tolerance among individuals of the two species as such variation might serve as a substrate for natural selection in warming seas.

## MATERIALS AND METHODS

### Collection site

Sea urchins of both *Diadema antillarum* and *Echinometra lucunter* were collected (under a permit granted by the Cayman Islands Department of Environment), by snorkeling in a small rectangular inlet south of George Town harbour, Grand Cayman, located at 19°16′42.69″N, 81°23′34.79″W, (in the Marine Protected Area between study sites GCM9 and GCM5 of *McCoy, Dromard & Turner (2009)*). The inlet was roughly 36 m long ×14 m wide with a gently sloped floor that reached the more open sea at a depth of about 3 m. The walls of the inlet were composed of eroded ironshore (*Logan, 2013*), with many cracks and crevices in which sea urchins, particularly *E. lucunter*, were found (Fig. 1A). The floor of the inlet was coral rubble and sand where patches of *D. antillarum* were found (Fig. 1B).

Both species were collected in the afternoon during the winter (February, 2013), spring (May, 2010) and summer (August, 2011) in order to assess the extent of acclimatization of thermal tolerance among animals that had likely been exposed to substantial inter-seasonal temperature variation. I measured the temperature at both shallow (10 cm below the surface) and deeper collection sites (2.5 m below the surface) during each collection visit using a Fisher Scientific Precision Thermometer, (accurate to 0.01 °C) or a mercury thermometer (accurate to 0.1 °C). Unexpectedly, there was no temperature difference with depth. I confirmed this by collecting temperatures at 15-min intervals for several days to a full week in both shallow and deeper collection sites (Onset HOBO temperature probes and data logger). The shallow site was not a sequestered intertidal pool but rather was flushed with seawater constantly, perhaps accounting for the absence of a temperature differential between shallow and deeper sites. Therefore, collection temperatures are reported as the combined mean from the shallow and deep sites for each season.

### Animal maintenance in the laboratory

Animals were maintained in a laboratory at the Cayman Islands Department of Environment building in George Town, Grand Cayman. Between 4 and 8 sea urchins of either species were maintained in a 50 l tank of aerated seawater at room temperature (22.5 ± 0.7 °C) and under natural light. All animals were tested within 3 days of collection. Animals were not fed while in the laboratory.

### Measures of thermal tolerance: righting time and temperature at loss of righting

I measured both the righting time as temperature was increased and the temperature at which each sea urchin could no longer right itself ($T_{Lor}$) after inversion as indicators of thermal tolerance. The latter measurement is repeatable and indicates the temperature at which the neuromuscular system of the animal no longer has sufficient integrity to mediate righting. It has been a reliable indicator of stress in many animal species including echinoids (*Lawrence & Cowell, 1996*; *Challener & McClintock, 2013*; *Ubaldo, Uy & Dy, 2007*).

Each sea urchin was tested individually and placed in a tank with a volume of room temperature aerated seawater sufficient to cover the urchin and provide room for me

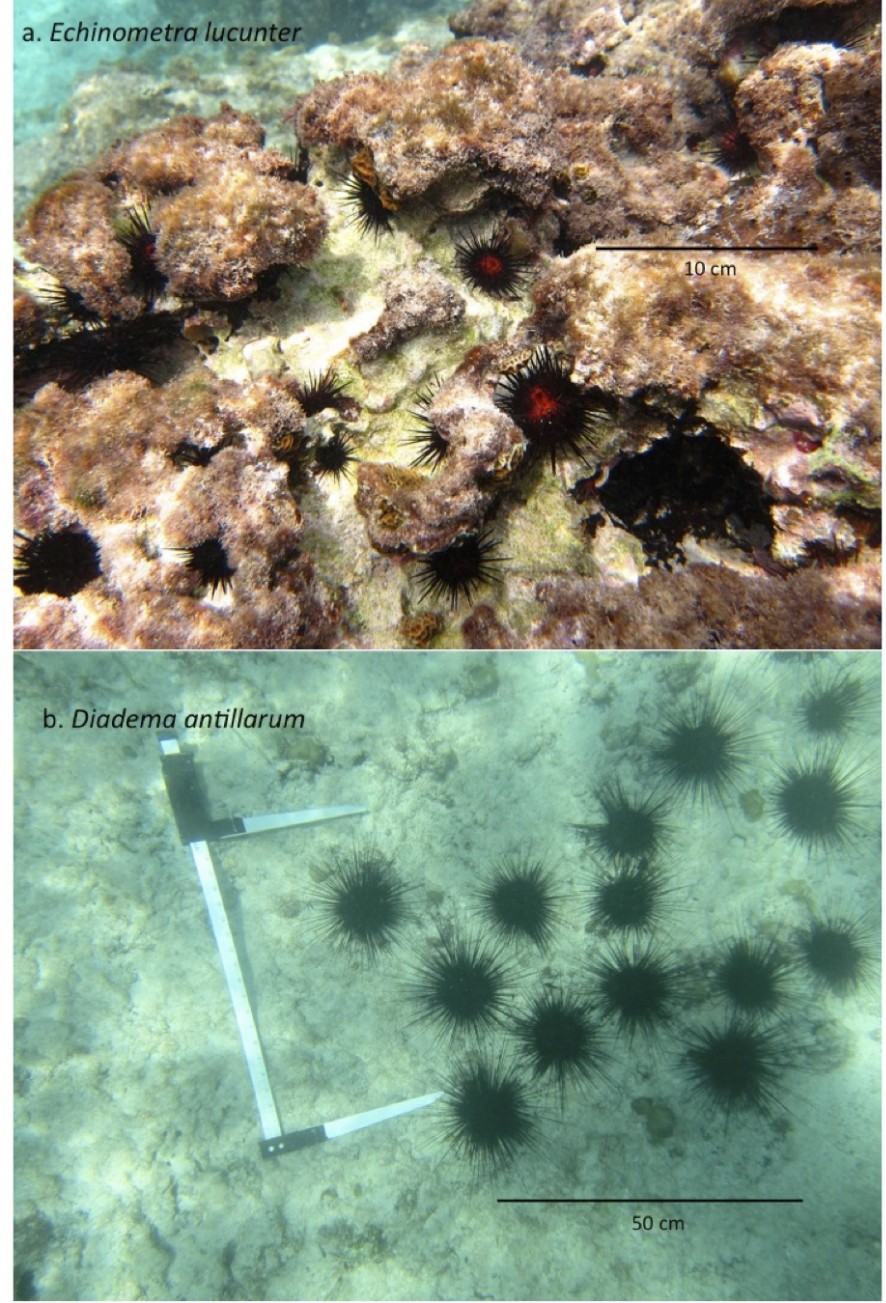

**Figure 1 Sea urchins in field sites.** (A) *Echinometra lucunter*. (B) *Diadema antillarum*.

to invert the animal. The experimental substrate was smooth without sand or gravel. The water was aerated through the entire experiment, which also mixed the water as it was heated. At the beginning of each trial (in room temperature water), the sea urchin was placed on its aboral surface (using wooden spatulas) and the time for the animal to right itself completely (its oral surface contacting and parallel to the tank bottom) was recorded. The testing tanks were large and could not be heated sufficiently using available

heaters. In order to heat the water, therefore, small volumes of test water were removed (without disturbing the test animal) and were replaced by comparable volumes of aerated and heated water from a different tank (with a smaller volume that was easier to heat). With practice following pilot studies, I was able to reliably increase the water temperature in the test tank at a rate of 0.16–0.3 °C min$^{-1}$. I constantly mixed the water in the test tank with a paddle and measured the water temperature with a Fisher Scientific Precision Thermometer, (accurate to 0.01 °C) which remained in the test tank. As the water was heated, the animal was inverted periodically (roughly once for each degree increase, and more frequently as loss of righting ability approached) and the righting time was recorded. The end point of the loss of righting was taken as the temperature at which the animal no longer could right itself (within 15 min of inversion), and was designated the $T_{LoR}$. The animals were not dead at loss of righting, as they continued to move their spines and mouthparts. Moreover, after the $T_{LoR}$ was noted, each animal was immediately placed in room temperature seawater after which coordinated movement was restored. The sea urchins did not appear to habituate to inversion as the righting times for an individual did not get progressively longer through the experiment until $T_{LoR}$ was imminent.

In order to determine if age (size) influenced the $T_{LoR}$, I measured the test diameter of each sea urchin following the experiment with vernier calipers (to the nearest mm). If the test shape of the sea urchin was not a circle, I measured the major axis. *D. antillarum* have spines of very different length while those of *E. lucunter* are more similar in length and this difference in morphology appeared to affect how the animals maneuvered themselves during righting. Therefore, I also measured the length of a spine that appeared to be among the longest on each animal.

In other studies (e.g. *Lawrence, 1973*), inverted animals were placed in water of particular temperatures and then righting time was measured, whereas in this study, the water was heated until the animal could no longer right itself. Heating rates affect the thermal tolerance of many marine invertebrates (*Nguyen et al., 2011*). The data in the present study reflect acute responses to increasing temperatures of animals taken from the wild that had been acclimatized to winter, spring, or summer temperatures.

The mean collection temperatures (±SEM) were 27.2 °C (±0.12) in February, 28.8 °C (±0.13) in May, and 30.4 °C (±0.09). In Fig. 2, I compared the mean collection temperatures of this study with data from the Cayman Islands provided by the National Oceanic and Atmospheric Administration (NOAA) *NOAA Coral Reef Watch* (*2000*, sampled twice weekly). I used NOAA data from 2010–2013 in the months of February, May and August to determine if my collection temperatures were representative of typical seasonal Cayman Islands reef water temperatures. In both data sets, there are comparable changes in mean temperatures with season. Moreover, the temperatures I measured are consistent with those reported in the NOAA data albeit slightly higher by 0.58–0.8 °C. I collected temperatures during the day while the NOAA data were collected at night, accounting for the slightly greater temperatures reported here (Fig. 2).

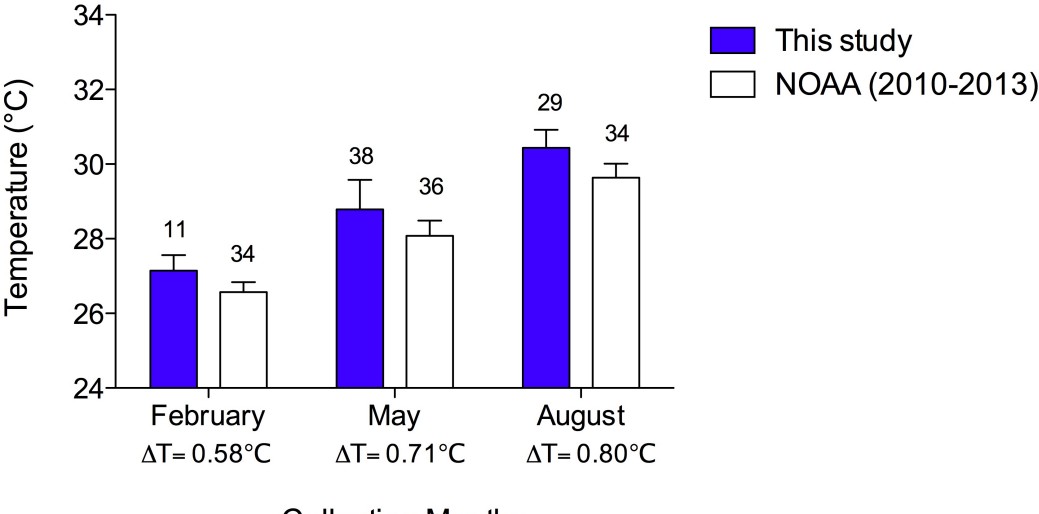

**Figure 2 Sea temperatures reported in this study compared to reef sea temperatures of the Cayman Islands in February, May, and August, 2010–2013, reported by *NOAA Coral Reef Watch*.** Bars represent means with vertical lines representing 1 SEM. Sample sizes are indicated above bars. The difference between mean temperature in this study and that of NOAA data are indicated by a $\Delta$ below the x-axis. A two-way ANOVA revealed that temperatures increased with month ($p < 0.0001$) in both data sets and that the NOAA data (recorded at night) were significantly lower than the data from the present study, which were recorded during the day ($p < 0.0001$). There was no interaction of temperature and study ($p = 0.541$).

## Statistical analyses

To determine if there were differences between species and among collection temperatures in loss of righting temperatures ($T_{LoR}$), I performed a two-way ANOVA with species and collection temperatures as fixed factors and $T_{LoR}$ as the dependent variable. At temperatures below the $T_{LoR}$ ($\leq 30.9$ °C) I analyzed whether species and collection temperatures affected righting times using a multiple regression analysis. I also compared the ratio of the length of a long spine to test diameter between the two species using the Student's t-test. Finally, I performed linear regression analyses to determine if size (test diameter) was a predictor of $T_{LoR}$ for each species in each season.

## RESULTS

The mean temperatures for loss of righting ($T_{LoR}$) of both species were greater than the collection temperatures and increased with increasing collection temperature, so that those collected in the winter had the lowest thermal tolerance and the summer collected sea urchins had the highest (Fig. 3). *D. antillarum* lost righting ability at significantly higher temperatures than *E. lucunter* at all collection temperatures (Fig. 3). The 2-way ANOVA showed that both species ($p < 0.0001$) and collection temperature ($p < 0.0001$) were significant predictors of $T_{LoR}$ but their interaction was not ($p = 0.12$). The difference between the mean collection temperature and the mean $T_{LoR}$, which can be thought of as the thermal safety margin (*Nguyen et al., 2011*), was more than 3 °C for *E. lucunter* in all seasons (Table 1). The thermal safety margin of *D. antillarum* was almost 6 °C in winter

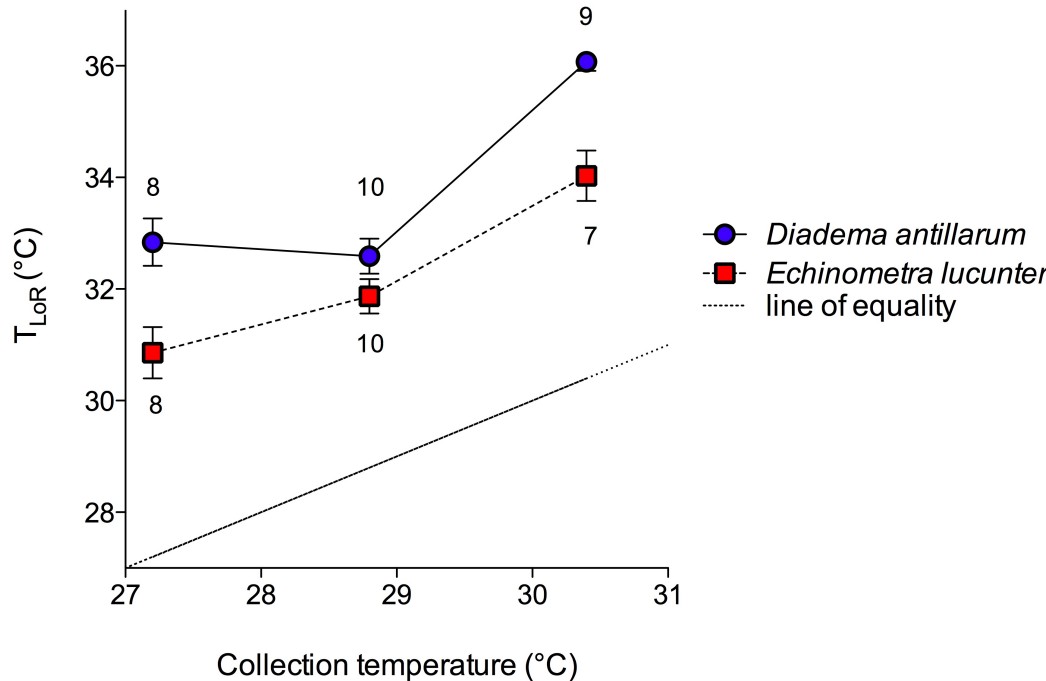

**Figure 3 Temperature at the loss of righting ($T_{LoR}$) of *D. antillarum* and *E. lucunter* at different collection temperatures.** The symbols represent mean $T_{LoR}$ with vertical lines representing 1 SEM. Sample sizes are indicated above or below the symbol. The dotted line represents the line of equality, $T_{LoR}$ = sea temperature. The thermal safety margin is the distance between the fitted lines and the line of equality. The main effects of species ($F_{1,46} = 29.51, p < 0.0001$) and collection temperature ($F_{2,46} = 48.17, p < 0.0001$) were significant predictors of $T_{LoR}$ but their interaction was not ($F_{2,46} = 2.189, p = 0.12$).

**Table 1 Difference between mean $T_{LoR}$ and mean collection temperature (thermal safety margin) for both species.**

|  | *D. antillarum* | *E. lucunter* |
|---|---|---|
| FEBRUARY | 5.64 °C | 3.66 °C |
| MAY | 3.79 °C | 3.07 °C |
| AUGUST | 5.67 °C | 3.63 °C |

and summer but only 3.79 °C in the spring. Thus, the $T_{LoR}$ of *E. lucunter* increased roughly linearly with seasonal temperature. However, the $T_{LoR}$ of *D. antillarum*, while greater than that of *E. lucunter*, did not change from winter to spring, although it was higher in the summer (Fig. 3; Table 1).

There was variation in $T_{LoR}$ within both species (Table 2). *Diadema* exhibited the largest range in $T_{LoR}$ during the winter (5.3 °C) while *Echinometra* exhibited the largest range in $T_{LoR}$ during the summer (3.6 °C).

The rock boring *E. lucunter* took longer to right themselves compared to *D. antillarum* at all temperatures below the $T_{LoR}$ ($p < 0.0001$; Fig. 4). Collection temperature also affected the righting times below the $T_{LoR}$ ($p = 0.025$), likely due to the slower righting

**Table 2 Ranges in $T_{LoR}$ of both species across the 3 collecting seasons.** Numbers in parentheses represent the difference between the highest and lowest $T_{LoR}$ for each species.

|  | D. antillarum | E. lucunter |
|---|---|---|
| FEBRUARY | 30.0–35.3 °C (5.3) | 29.9–32.3 °C (2.4) |
| MAY | 31.3–34.2 °C (2.9) | 30.3–33.4 °C (3.1) |
| AUGUST | 35.2–36.7 °C (1.5) | 32.0–35.6 °C (3.6) |

times of winter collected *Diadema*. The temperature at which the sea urchins were tested did not have a significant effect on righting times ($p = 0.561$). Typically, the *E. lucunter* that righted themselves took between 100 and 240 s while *D. antillarum* righted themselves in less than 100 s. I have provided video examples of the righting behavior of both species in the Supplemental Information.

When the long-spined sea urchin righted itself, it appeared to rely on the levering action of the longer spines, which exceeded the diameter of the test, to rock the animal onto its side. The rock boring sea urchin, on the other hand, used its tube feet rather than its spines, which were shorter than the test diameter, to push and pull itself onto its side. The greater mean ratio of the length of a spine to test diameter in *Diadema* ($1.38 \pm 0.1$) compared to *Echinometra* ($0.78 \pm 0.04$) is illustrated in Fig. 5 ($p < 0.0001$).

$T_{LoR}$ did not vary with test diameter in either species (Fig. 6; for *D. antillarum*, $p$ values for all seasons $>0.4$; for *E. lucunter*, $p$ values for all seasons $>0.1$). Although the test diameters overlapped between the two species, on average *Diadema* were larger than *Echinometra*.

## DISCUSSION

Temperature affects the physiological responses of tropical marine animals more than other possible stressors of climate change (*Przeslawski et al., 2008*). Both species of sea urchin in this study exhibited acclimatization to seasonal temperature changes in experiments measuring acute thermal tolerance (Fig. 3), their $T_{LoR}$ increasing with increases in environmental temperature. Nevertheless, like other tropical animals, the safety margin of thermal tolerance above ambient temperatures ($\sim$3–6 °C, Table 1) was small compared to that of temperate animals (*Przeslawski et al., 2008*; *Nguyen et al., 2011*). It is possible that the thermal safety margins of the sea urchins would be different with a slower rather than acute temperature increase that would more closely resemble natural temperature changes. However, *Nguyen et al. (2011)* demonstrated that subtidal tropical animals exhibited lower thermal tolerances when heated at slower rates. So the thermal tolerances reported in this study likely represent the highest tolerances that these animals can exhibit given how rapidly they were heated. Nevertheless, different thermal tolerance end points and different heating rates (e.g., *Hernández et al., 2004*; *Nguyen et al., 2011*) make comparisons of particular thermal tolerances across different studies difficult.

*Diadema* exhibited the largest range in $T_{LoR}$ during the winter and the smallest range during the summer, while *Echinometra* showed the opposite tendency, with the largest

A.

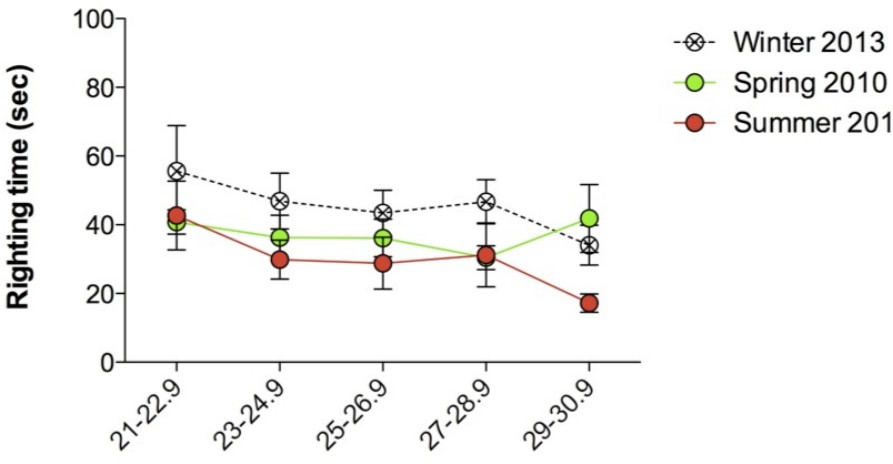

B.

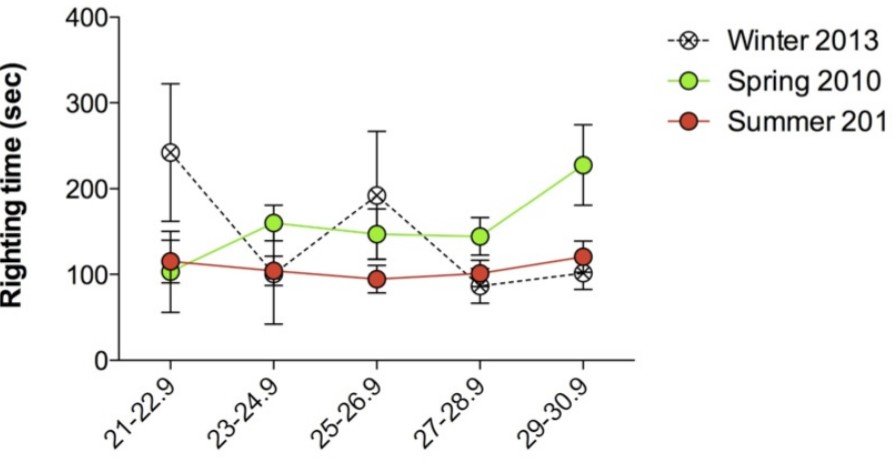

**Figure 4 Mean righting times (±SEM) at increasing experimental temperatures (below $T_{\mathrm{LoR}}$) of (A) *Diadema* and (B) *Echinometra* collected during the winter, spring, and summer.** For clarity, the data are displayed in bins of experimental temperatures of 1.9 °C. Species ($p < 0.0001$) and collection temperature ($p = 0.025$) significantly affected righting time but the temperatures at which the sea urchins were tested did not ($p = 0.561$). Model: $F_{3,431} = 56.58$; $p < 0.0001$; $R^2 = 0.28$. Note that the $y$-axes are scaled differently.
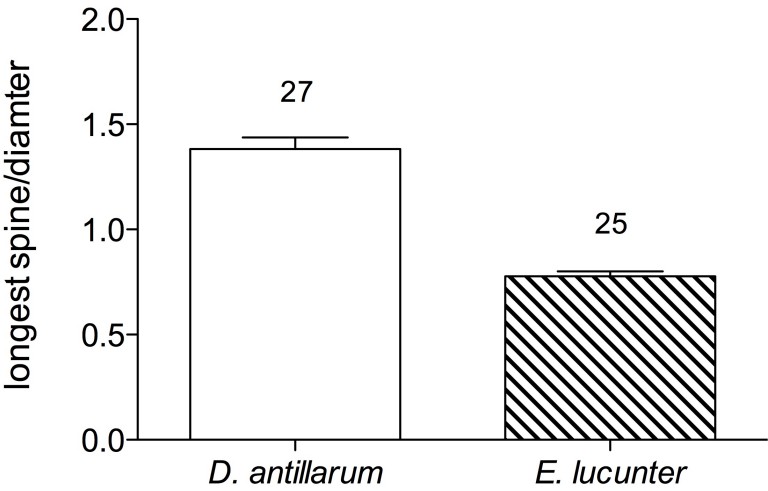

**Figure 5 Mean ratio of longest spine per test diameter (±SEM) of *Diadema* and *Echinometra* ($t = 10.74, df = 51; p < 0.0001$).** Sample sizes are indicated above the bars.

range in $T_{LoR}$ in the summer (Table 2). These variations in thermal tolerance suggest some capacity for evolutionary adaptation to temperature changes if the variation among adults has a genetic component. The sea urchins also exhibited variation in righting times at temperatures below the $T_{LoR}$ (Fig. 4) and this variation may affect their daily movements such as foraging, in warming seas. Like other benthic invertebrates, sea urchins have pelagic larvae, which may be exposed to different conditions from those of adults. Since larvae disperse widely (*Lessios, 1988*), urchins have the potential to establish in new areas, facilitating range changes in response to global climate change (*Ling et al., 2009*; *Przeslawski et al., 2008*). Alternatively, if sea urchins cannot adapt to the rapid changes in their environments, local populations may become extinct.

The proximate causes of interspecific differences in thermal tolerance responses are unclear (Figs. 3 and 4). Both species were exposed to similar daily and seasonal temperature variations. One possible proximate cause of the faster righting time of *Diadema* is the difference in spine length relative to test size (Fig. 5). The *Diadema* used their relatively long aboral spines as levers to topple them over when righting. The *Echinometra*, having much shorter spines, used their tube feet to right themselves, which was a more time-consuming process. The different uses of spines versus tube feet have been reported in other motor behaviors of sea urchins as well by *Domenici, González-Calderón & Ferrari (2003)*. They, however, found that size affected motor behavior on vertical and horizontal surfaces, whereas size had no effect on righting behavior in my study (Fig. 6), nor did it affect thermal safety margins in other tropical ectotherms (*Nguyen et al., 2011*). The different relative length of spines might explain why *Echinometra* took longer to right themselves at temperatures below $T_{LoR}$. However, it would not account for the lower $T_{LoR}$ (Fig. 3) and smaller thermal safety margin (Table 1) of *Echinometra*. Righting behavior may have more survival value to *Diadema* than *Echinometra*. Both species forage at night (*Hendler et al., 1995*) but only *Diadema* were found in the open during the day and I have

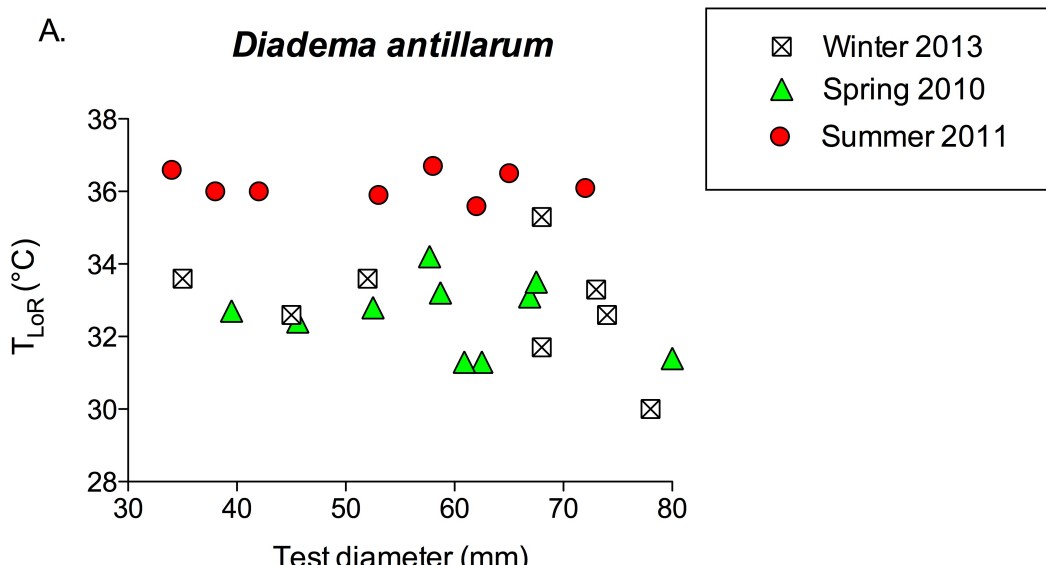

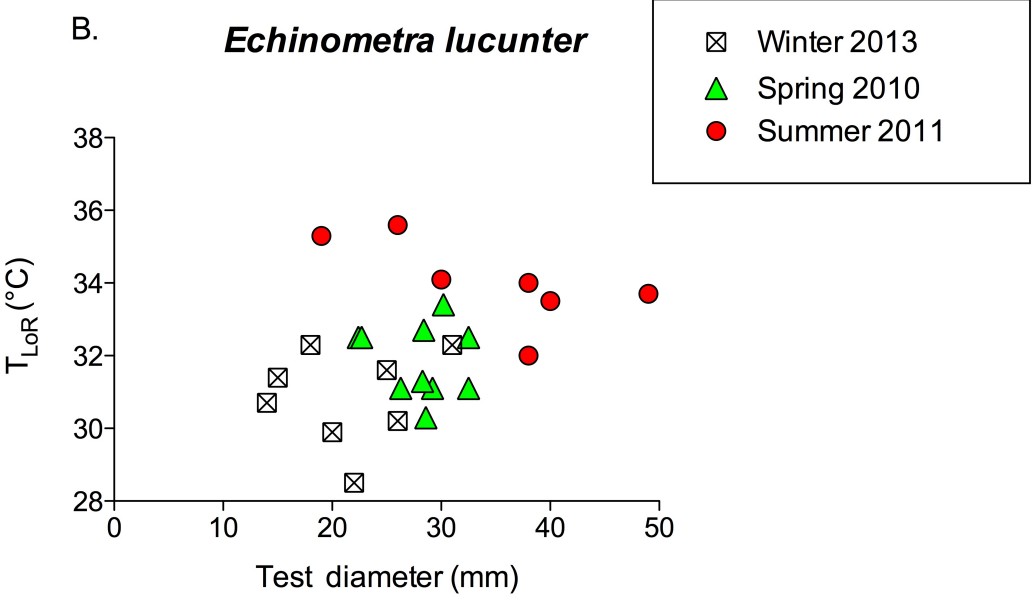

**Figure 6** $T_{LoR}$ **as a function of test size in (A)** *Diadema* **and (B)** *Echinometra* **collected in the winter, spring, and summer.** None of the slopes is significantly different from 0. *Diadema*: $p_{winter} = 0.396$; $p_{spring} = 0.528$; $p_{summer} = 0.818$. *Echinometra*: $p_{winter} = 0.783$; $p_{spring} = 0.689$ $p_{summer} = 0.10$. Note that the *x*-axes are scaled differently.

observed fish flipping them over in order to eat them. The rock-boring urchins, on the other hand, as the name implies, are jammed into crevices and typically not found in the open during the day. At night, the temperature can fall by as much as 1 °C in the summer (E Sherman, 2011, unpublished data) so *Echinometra* are less likely to be exposed and vulnerable during higher daytime temperatures.

Phase shifts from coral to macroalgal dominance following the *Diadema antillarum* die off in the Caribbean have been widely reported (e.g., *Jackson, 2001*; *Lessios, 1988*). There appears to be great regional variation in the recovery of different populations of these sea urchins (*Carpenter & Edmunds, 2006*; *Levitan, Edmunds & Levitan, 2014*; *Miller et al., 2009*). Regions in which *Diadema* are increasing in number are correlated with increases in coral settling and cover (*Carpenter & Edmunds, 2006*; *Knowlton, 2001*; *Furman & Heck Jr, 2009*; *Idjadi, Haring & Precht, 2010*). Unlike *D. antillarum* populations, *E. lucunter* populations have been relatively stable throughout the Caribbean and have not been associated with changes in coral density (*Furman & Heck Jr, 2009*). Both species of sea urchins in the present study were collected from local populations in a small area in a Grand Cayman reef. It is not clear, therefore, how representative their thermal tolerance responses are compared to sea urchins from other areas. Both species demonstrated a significant capacity to acclimatize to different seasonal temperatures, which may serve them well as sea temperatures increase. *Nguyen et al. (2011)* suggested that temperature increases as low as 2–3 °C above present levels are likely to be stressful to tropical marine ectotherms. The sea urchins in this study, however, appear to be able to tolerate such changes. Additional studies are required to determine the capacity of sea urchins to adapt to acute versus long-term thermal stress and the costs of such adaptation as global sea temperatures increase.

## CONCLUSION

Tropical sea temperatures may increase by as much as 4.8 °C by the end of this century (*IPCC, 2014*). It remains to be seen if sea urchins will be able to adapt to these higher temperatures. Moreover, other stressors (such as ocean acidification, infectious disease, runoff from land) are likely to have synergistically deleterious effects on tropical marine ectotherms (*Przeslawski et al., 2008*). Populations may adapt to higher sea temperatures by natural selection given the within season variation in thermal tolerance exhibited by sea urchins in this study. The sea urchins revealed at least seasonal plasticity in their capacity to acclimatize to different temperatures. Alternatively, if sea temperatures increase more rapidly than can be accommodated by sea urchins, local populations may become extinct. Changes in the number of *Diadema antillarum*, in particular, will have important consequences for the structure of coral reefs.

## ACKNOWLEDGEMENTS

This work was supported in large part by the Cayman Islands Department of Environment, which supplied lab space, technical assistance, and sea urchin collecting permits. I am grateful to this fine institution directed by Gina Ebanks-Petrie. Particular thanks go to staff scientists Tim Austin, John Bothwell, Gene Parsons, and James Gibb. Nancy Knowlton and Jeremy Jackson discussed this work with me and I appreciate their insights. Thanks

to Kathryn Montavan for help with statistical analyses. Kerry Woods provided valuable comments on a prior version of this manuscript. I am grateful to the PeerJ reviewers whose suggestions and comments enhanced the quality of this paper. I would also like to thank Dusty Norman of DNS Diving for technical assistance. Finally, thanks go to the people who helped collect sea urchins: Amie McClellan, Terry Creach, Katie Alpers.

### Funding

This study was funded by grants from Bennington College. The funders had no role in study design, data collection and analysis, decision to publish, or preparation of the manuscript.

### Grant Disclosures

The following grant information was disclosed by the author:
Bennington College.

### Competing Interests

The author declares there are no competing interests.

### Author Contributions

- Elizabeth Sherman conceived and designed the experiments, performed the experiments, analyzed the data, contributed reagents/materials/analysis tools, wrote the paper, prepared figures and/or tables, reviewed drafts of the paper, photos and videos.

### Field Study Permissions

The following information was supplied relating to field study approvals (i.e., approving body and any reference numbers):

Sea urchins were collected under a permit from the Cayman Islands Department of Environment. There was no number associated with the permit.

### Supplemental Information

Supplemental information for this article can be found online at http://dx.doi.org/10.7717/peerj.1006#supplemental-information.

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
