# Peer review of "Can sea urchins beat the heat? Sea urchins, thermal tolerance and climate change"

_PeerJ, doi:10.7717/peerj.1006_

## Round 0.1 · original submission · Major Revisions

· Academic Editor

Major Revisions

Dear Dr Sherman,

Thank you for submitting your manuscript to PeerJ. Please note that the reviews have highlighted several points which need to be addressed.

Reviewer 1 ·

Basic reporting

This is an interesting paper on thermal limits in sea urchins but it misses the mark in several respects. It starts with reporting a massive die off but never explains it. It uses its own definition of CTmax as if that was ecologically relevant but it may well be more extreme than the earliest thermal constraints felt by the organisms in the field. The OCLTT concept has identified these limits as the onset of performance reductions whereas the present paper quantifies the temperature of full loss only. It would be worth addressing this and bring the paper up to speed with respect to available concepts.

Abstract: The definition of CTmax does not match the original definition which has more severe phenomena (spasms) than the loss of righting response. Authors should consider using a different term to avoid confusion.

l. 24: Munday et al. 2009 ist probably not the authoritative reference to cite here but the recent IPCC report with regional projections. There in the ocean chapters and relevant cross chapter boxes the author will find the relevant background information.

l. 29-30 This sentence makes no sense. Would you then expect different results?

l. 101: the rate of heating may be sufficient for temperature equalization but there is a time dependence of CTmax. Values may result lower during slower warming as acknowledged later. This should be discussed in light of finding in other invertebrates.
l. 102: can only guess what internal vs external means.

l. 113 to 114: One wonders about the temperature dependence of this process. It needs to be made clear whether the rate of righting changes with temperature?

l. 117: awkward sentence
l. 125: time dependence of CTmax needs to be considered as acknowledged on l. 180 to 184 and should clearly have been investigated.

Experimental design

conceptual limits addressed in basic reporting

Validity of the findings

conceptual limits addressed in basic reporting, field limitations and limiting temperatures operative in the field have not been addressed.

Additional comments

see basic reporting

·

Basic reporting

In this manuscript, the author compares the critical thermal maxima (CTmax) of two species of urchins over different seasons. They find that CTmax varies both between species and among seasons, and that the difference between CTmax and environmental temperatures tend to be low so thermal safety margins are small relative to temperate species. The manuscript is well-written and clear, albeit a bit sparse on data. It also helps address the relative paucity of thermal sensitivity data on echinoids.

Minor comments:

Methods and results: please indicate in the text how many individuals were tested throughout.
Figure 2. Please indicate significant differences using letters or symbols. Also, it’s not entirely clear from the plot or caption how long or many temperature measurements were taken—is this all from during collection, so the variation represents microhabitat variation at the same time point?
Figure 3. Please indicate significant differences using letters or symbols.
Figure 5. Please indicate significant differences using letters or symbols.
Figure 6. Suggest adding results of statistical test to the caption. Suggest also placing both figures on the same x-axis to show differences in test size between species.

Experimental design

The experimental design is generally clear, but there is one major confound. Since the urchins were collected during different years, what evidence do you have that indicates the years were similar? Do you think differing temperature regimes during different years may have affected your results?

Other comments:
1.. What seawater temperature data is available? Do temperatures vary enough that these CTmax values may be exceeded at any point in the year? Seeing some of the mentioned HOBO data would also help place some of these CTmax values in context.
2. Why not test differences in righting time between species and among collection temperatures?
3. Line 101: How was the water heated?
4. It looks like there may be a relationship between test size and CTmax in E. lucunter, but that the sample size was insufficient to test it. Have you conducted power analysis to investigate this possibility?

Validity of the findings

I disagree with the author’s conclusion that the small thermal safety margin indicates sensitivity to ocean warming—since they demonstrate that urchins are capable of acclimation, it is possible that plasticity may mitigate the impacts of ocean warming in these species. Additional experiments showing a limit to or a cost to plasticity, would be needed for this conclusion to be drawn. I suggest some discussion of the literature on plasticity and climate change might strengthen the manuscript (e.g. Williams et al. 2008, PLoS Biol.)

Additional comments

I find the manuscript is generally clear and adds to the literature on thermal tolerance of marine invertebrates. I think additional discussion of seawater temperature variability from year to year is necessary (showing in particular similarities and differences in temperature regime among the collection years), as well as some discussion of the role of phenotypic plasticity for potentially mitigating climate change effects. Finally, I think the manuscript would be strengthened by additional statistical investigations of the differences in righting time between species and collection temperature."

---

## Round 0.2 · Minor Revisions

· Academic Editor

Minor Revisions

Dear Dr Sherman,

Thank you for submitting your corrected version of the manuscript. The reviewers are happy that they suggestions/corrections have been addressed. I have minor minor correction I would like you too consider prior to acceptance.

The first line of your conclusion is a little strong "Sea urchins will likely encounter stressful temperatures in the next 100 years due to global climate change" and I don't think the way it is worded is backed up by the the current experiment. I would urge a more measured conclusion taking into account adaptation and the IPCCs range predictions for the next 100 years. For example....Sea urchins will likely encounter stressful temperatures IF THEY ARE UNABLE TO ADAPT TO THE PREDICTED RISES IN TEMP OVER....in the next 100 years due to global climate change.

Best wishes,

Alex Ford

·

Basic reporting

The MS is well and clearly written, with a nice description of the study system and how the results fit into the broader ecological picture of the area.

Experimental design

The experimental design is thoughtful and clearly laid out, and enough details are given to be reproducible by other investigators.

Validity of the findings

The data are clear, and well presented. The author has taken the time to review and revise their conclusions, which I feel strengthens the MS.

Additional comments

In this revised version of the MS, it appears the author has taken my previous comments seriously, and spent a considerable amount of time carefully revising their manuscript.

---

## Round 0.3 · accepted · Accept

· Academic Editor

Accept

Dear Dr Sherman,

Thank you once again for submitting to PeerJ and making those additional amendments to the manuscript.